# A Novel Hybrid Approach for Classifying Osteosarcoma Using Deep Feature Extraction and Multilayer Perceptron

**DOI:** 10.3390/diagnostics13122106

**Published:** 2023-06-18

**Authors:** Md. Tarek Aziz, S. M. Hasan Mahmud, Md. Fazla Elahe, Hosney Jahan, Md Habibur Rahman, Dip Nandi, Lassaad K. Smirani, Kawsar Ahmed, Francis M. Bui, Mohammad Ali Moni

**Affiliations:** 1Centre for Advanced Machine Learning and Applications (CAMLAs), Bashundhara R/A, Dhaka 1229, Bangladesh; tarekiub17@gmail.com (M.T.A.); elahe.se@daffodilvarsity.edu.bd (M.F.E.); jahan@cse.mist.ac.bd (H.J.); habib@iu.ac.bd (M.H.R.); 2Department of Computer Science, American International University-Bangladesh (AIUB), 408/1, Kuratoli, Khilkhet, Dhaka 1229, Bangladesh; dip.nandi@aiub.edu; 3Department of Software Engineering, Daffodil International University, Daffodil Smart City (DSC), Savar, Dhaka 1216, Bangladesh; 4Department of Computer Science & Engineering (CSE), Military Institute of Science and Technology (MIST), Mirpur Cantonment, Dhaka 1216, Bangladesh; 5Department of Computer Science and Engineering, Islamic University, Kushtia 7003, Bangladesh; 6The Deanship of Information Technology and E-learning, Umm Al-Qura University, Mecca 24382, Saudi Arabia; lksmirani@uqu.edu.sa; 7Department of Electrical and Computer Engineering, University of Saskatchewan, 57 Campus Drive, Saskatoon, SK S7N 5A9, Canada; francis.bui@usask.ca; 8Group of Biophotomatiχ, Department of Information and Communication Technology (ICT), Mawlana Bhashani Science and Technology University (MBSTU), Tangail 1902, Bangladesh; 9Artificial Intelligence & Digital Health, School of Health and Rehabilitation Sciences, Faculty of Health and Behavioural Sciences, The University of Queensland, St. Lucia, QLD 4072, Australia; m.moni@uq.edu.au

**Keywords:** osteosarcoma, convolutional neural networks, transfer learning, feature extraction, feature selection, machine learning, MLP

## Abstract

Osteosarcoma is the most common type of bone cancer that tends to occur in teenagers and young adults. Due to crowded context, inter-class similarity, inter-class variation, and noise in H&E-stained (hematoxylin and eosin stain) histology tissue, pathologists frequently face difficulty in osteosarcoma tumor classification. In this paper, we introduced a hybrid framework for improving the efficiency of three types of osteosarcoma tumor (nontumor, necrosis, and viable tumor) classification by merging different types of CNN-based architectures with a multilayer perceptron (MLP) algorithm on the WSI (whole slide images) dataset. We performed various kinds of preprocessing on the WSI images. Then, five pre-trained CNN models were trained with multiple parameter settings to extract insightful features via transfer learning, where convolution combined with pooling was utilized as a feature extractor. For feature selection, a decision tree-based RFE was designed to recursively eliminate less significant features to improve the model generalization performance for accurate prediction. Here, a decision tree was used as an estimator to select the different features. Finally, a modified MLP classifier was employed to classify binary and multiclass types of osteosarcoma under the five-fold CV to assess the robustness of our proposed hybrid model. Moreover, the feature selection criteria were analyzed to select the optimal one based on their execution time and accuracy. The proposed model achieved an accuracy of 95.2% for multiclass classification and 99.4% for binary classification. Experimental findings indicate that our proposed model significantly outperforms existing methods; therefore, this model could be applicable to support doctors in osteosarcoma diagnosis in clinics. In addition, our proposed model is integrated into a web application using the FastAPI web framework to provide a real-time prediction.

## 1. Introduction

Osteosarcoma, also known as osteogenic sarcoma, is a primary mesenchymal tumor that is distinguished histologically by the formation of osteoid by malignant cells. Osteosarcoma affects people mostly between the ages of 10 and 30, making it the third most common cancer among children and adolescents. The United States reports approximately 1000 new cases every year [1,2] that illustrate osteosarcoma as a challenging issue. Osteosarcoma and Ewing sarcoma are the two malignant bone cancers that mostly affect children and adolescents, and they represents about 56% and 34% bone cancer, respectively. The most common sites for osteosarcoma are the femur (42%, 75% of which are in the distal femur), the tibia (19%, 80% of which are in the proximal tibia), and the humerus (10%, 90% of which are in the proximal humerus) [3]. Osteosarcoma signs typically start off as mild localized bone pain, warmth, and redness where the tumor is located [4]. Neoadjuvant chemotherapy (NAC) and surgery are current therapeutic modalities that have improved patient survival rates by almost five years. From 1975 to 2010, osteosarcoma patients experienced an increase in five-year survival rate from 40% to 76% for those under the age of 15 and from 56% to 66% for those aged between 15 and 19 [5]. However, the five-year survival rate for metastatic osteosarcoma is still under 20% [6]. Early osteosarcoma diagnosis and careful monitoring during the chemotherapy cycle can increase the overall survival rate [7].

For the majority of malignancies, including osteosarcoma, a biopsy (histology report) test is the best way to determine if a part of the body has cancer. In addition, non-invasive imaging methods such as MRI, CT, and PET imaging modalities have been used for quantitative analyses in osteosarcoma response monitoring and surgical planning [7]. Even though the approaches based on biopsy can successfully identify the malignancy, approaches such as histologically guided biopsies and similar techniques have limitations in detecting malignancy. Moreover, the process of preparing histology specimens takes time; e.g., to represent the surface of a substantial three-dimensional tumor at least 50 histology slides are required to detect osteosarcoma malignancy accurately [8]. While assessing cancer patients using biopsy-derived tissue slides, pathologists manually find the most affected areas and examine the nuclear morphology and cellular characteristics. This manual inspection and diagnosis using tissue slides, which may consist of huge number of cells, can be laborious and arbitrary. The whole slide image (WSI) analysis can increase the amount of information retrieved from tissue slides for making the decision and increase the reliability of analysis [9]. An automated method is expected to emerge for the histopathological slide classification of osteosarcoma because microscopic analysis of slides is difficult, time-consuming, tedious, and subject to bias [10]. The morphological and contextual cues present in the digital WSIs are used as features for tissue classification, which promotes the usage of image processing and analysis approaches [9,11].

Osteosarcoma is highly heterogeneous, and it is influenced by inter- and intra-observer differences. In osteosarcoma, the precursor cells and some types of tumor cells are both stained the same shade of blue, but the precursor cells are rounder, more closely spaced, and more regular than the tumor cells [12,13]. To accurately determine the percentage of necrosis, various histological regions must be taken into account, including hemorrhagic tumor, blood cells, growth plates, clusters of nuclei, fibrous tissues, osteoclasts, cartilage, osteoid, osteoblasts, and precursors [10]. Recent research based on medical data shows that CNN can be used for medical images to extract and analyze information [14,15] and has a very successful impact. Deep learning (DL) and machine learning (ML) methods achieved tremendous success and popularity in medical research for cancer classification. In this study, we propose a hybrid approach, combining CNN and ML algorithms to classify osteosarcoma malignancy by using whole slide images for three classes, specifically, a viable tumor, necrosis (including fibrosis, osteoid, and coagulative necrosis), and nontumor (cartilage, bone, other normal tissue). The key contribution of our proposed hybrid approach is the integration of different CNN (transfer learning) and ML techniques for data preprocessing, optimizer analysis, feature selection, and classification for lower computational costs and better performance. Most of the previously published works either employed ML techniques or DL techniques where they only focused on classification tasks and did not explore enough other ML techniques that may influence accurate prediction. Initially, we performed normalization in the training phase to enable our model to learn faster with a zero-centered Gaussian distribution of data. We have also explored a total of five CNN models and five ML classifiers with different parameter settings to determine the best feature extractor and classifier integration. Furthermore, an optimizer analysis was conducted for the MLP classifier to ensure better optimization by selecting the most suitable optimizer that demonstrates improved convergence time and loss. We also analyzed the impact of RFE’s criterion concerning the number of selected important features. Finally, a web application has been developed using our proposed framework for real-time prediction.

The remaining sections of this article are structured as follows: In Section 2, we provided a literature review of previous works that has been published on this dataset. In Section 3, we described our proposed methodology in detail, including the dataset, preprocessing feature extraction, selection, and classification. The experimental results from various experiments, with proper analysis, are illustrated in Section 4. Furthermore, finally, we added a discussion in Section 5.

## 2. Literature Review

Researchers have developed automatic systems to identify various malignancies and tumors that can evaluate and classify medical images such as X-rays, histology images, ultrasound imaging, CT scan, MRIs, etc. [16,17,18,19,20,21,22,23,24,25]. The use of digital histopathology has grown significantly and shown great potential recently. In 2014, Irshad et al. [11] presented a survey on histopathology images, specifically in H&E and immunohistochemical staining protocols, that discusses classification techniques, segmentation, feature computation, and the major trends of various nuclei detection. Their study involves techniques including image thresholding, morphological features, active contour models (ACMs), K-means clustering, and probabilistic models. In order to distinguish between different tumor regions on osteosarcoma histology slides, Arunachalam et al. [26] demonstrated multi-level Otsu thresholding and shape segmentation. Mandava et al. [7] proposed an automatic segmentation technique of osteosarcoma using MRI images. A dynamic clustering algorithm called DCHS was proposed in their work, and it is based on a combination of fuzzy c-means (FCM) and Harmony Search (HS). To designate the tumor volume by DCHS, they used pixel intensity values and a subset of Haralick texture features as feature space. Nasor et al. [27] presented an automatic segmentation technique for osteosarcoma using MRI images combined with image processing techniques that includes K-means clustering, iterative Gaussian filtering, Chan–Vese segmentation, and Canny edge detection. An enhanced graph-cut-based framework was introduced by Vandana et al. [28] to determine malignancy level in H&E-stained histopathology images. They used mathematical morphology, color-based clustering, and active contour for extracting feature, and analyzed those features for malignancy classification using a multiclass random forest (RF) classifier. Zhi et al. [6] proposed ML approaches to classify osteosarcoma patients using metabolomics data analysis. LR, RF, and SVM are applied in their studies to distinguish between tumor cases and healthy controls. Feng et al. [29] presented a four pseudogene classifier to identify prognostic pseudogene signatures of osteosarcoma using RNA-seq data. The cox-regression analysis was used to construct the signature model (univariate, multivariate, and lasso), and achieved 0.878 AUC value.

Due to the availability of enormous computing power, DL approaches have gradually taken the place of traditional histopathological image classification [14,15,30,31,32]. In order to increase effectiveness and accuracy of osteosarcoma classification, Mishra et al. [10] developed a convolutional neural network (CNN). They have used WSI in their work to classify tumor classes (necrosis, viable tumor) versus nontumor class. The accuracy of their proposed CNN model was 92%, and the model was compared with three existing CNN models AlexNet, LeNet, and VGGNet. The first fully automated tool to evaluate viable and necrotic tumors in osteosarcoma is reported by Arunachalam et al. [33] that uses both DL and conventional ML techniques. Their intention was to classify the various tissue regions as viable tumor, necrotic tumor, or nontumor. They selected 13 different ML models in their study. Among them, the support SVM was the top performer, and a DL model was also developed to train on the same dataset. SVM, ensemble learner, and complex trees achieved an overall accuracy of 80.9%, 86.8%, and 89.9% respectively, and the overall accuracy for the deep learning model was 93.3% and 91.2% for patches and tiles of WSI’s. Osteosarcoma classification using histopathological images using sequential region-based convolutional neural network (R-CNN) was proposed by Nabid et al. [5] that consisted of bidirectional gated recurrent units (GRU) and CNN. Performance of their proposed model compared with AlexNet, SVM models, ResNet50, LeNet, and VGG16 on the same dataset and shows an accuracy of 89%. D’Acunto et al. [34] applied a DL approach to discriminate between Mesenchymal Stromal Cells (MSCs) from osteosarcoma cells and to classify the cell populations. A faster R-CNN was adopted in their study via transfer learning. A deep Siamese network model (DS-Net) was designed by Yu et al. [35] to develop an automated system for identifying viable and necrotic tumor regions in osteosarcoma. DS-Net was developed using a fully connected convolutional network that is combined with an auxiliary supervision network (ASN) and a classification network. Their model achieved an average accuracy of 95.1%. In order to find best classifier and to identify necrotic images from non-necrotic tissues, Anisuzzaman et al. [4] adopted six well-known pre-trained transfer learning CNN models. In their study, they employed both multiclass and binary class classification, and among the six pre-trained models, VGG-19 achieved the highest accuracy of 96%. Recently, S. Gawade et al.[36] employed multiple supervised deep-learning models to classify osteosarcoma, where they utilized a transfer learning approach that modifies only the top layer (classifier) and achieved the highest accuracy of 90.36% using ResNet. A comparative methodological approach was proposed by I.A. Vezakis et al. [37] to investigate different deep learning models. They considered various pre-trained models with transfer learning to perform normalization and resize input images into different sizes based on individual model sizes and obtained the highest accuracy of 91.00% for the MobileNetV2 model.

In recent years, ML based images processing approaches attracted a lot of interest and achieved a great success in the analysis of histopathological images of osteosarcoma. The literature survey motivated us to develop a hybrid model, combining DL and ML, to classify osteosarcoma using whole slide images. Firstly, a preprocessing technique was applied to the WSI cancer dataset to make the dataset more accurate format for analysis by the proposed method. We trained five cutting-edge CNN models to extract important features via transfer learning into a combined form of convolution and pooling from histopathological images. A decision tree-based RFE was developed to select the optimal number of features (e.g., 100, 200, …, 900) using a decision tree estimator from 1024 extracted features. Then, a modified MLP classifier was combined with different feature extractors with varying parameter settings for accurate prediction. Finally, we integrate the best data preprocessing, feature extractor, feature selector, and classifier to build our proposed model for predicting osteosarcoma. Here, we considered transfer learning with different hyperparameters that minimize the training time and provide more meaningful features. Moreover, feature selection techniques remove irrelevant features, thus reducing model complexity, and the modified classifier offers us more accurate classification results.

## 3. Methodology

This section describes dataset collection, image preprocessing, feature extraction, feature selection, and our proposed model. Figure 1 presents the schematic diagram of our proposed methodology.

The system Algorithm 1 of our proposed methodology is given as follows:
**Algorithm 1** Proposed Algorithm1:$E_{p}\leftarrow \mathrm{Number}\;\;\mathrm{of}\;\;\mathrm{Epochs}\;$2:W←TransferLearningModelParameter3:η←LearningRate4:$b_{s}\leftarrow \mathrm{Batch}\;\;\mathrm{Size}$5:D←OsteosarcomaDatasetOutput:Theassessmentmetricsonthetestdataset.DatasetPrepossessing:6:$X_{train}\leftarrow prepossessing\left(D\right)$7:$X_{test}\leftarrow prepossessing\left(D\right)$8:InitialiseTLModels(VGG16,VGG19,ResNet50Xception,DenseNet121)FeatureExtraction:9:**for **$\mathrm{local}\;\mathrm{epoch}\;e_{p}\;\leftarrow \;\mathrm{from}\;1\;\mathrm{to}\;E_{p}$** do**10:   **for** $b_{s}=\left(x_{s},y_{s}\right)\in \mathrm{random}\;\mathrm{batch}\;\mathrm{from}\;X_{train}$ **do**11:     Optimisemodelparameters12:     $W_{s}\leftarrow W_{s}-\eta \left(\Delta \left(L\left(W_{s};b_{s}\right)\right)\right)$13:     $f_{train}\leftarrow ComputeFeatures\left(W_{s},X_{train},1024\right)$14:   **end for**15:**end for**Feature Selection:16:$f_{best}\leftarrow DT-RFE\left(f_{train},900\right)$Osteosarcoma Tumor Classification:17:$TrainedModel\leftarrow MLP(f_{best},y_{train})$18:$Pred\leftarrow TrainedModel(X_{test})$19:$Evaluation\;metrics\leftarrow ComputeMetrics(Pred,y_{test})$

Initially, the input whole slide images (WSI) were pre-processed as described in Section 3.2 and then fed into the feature extractor. We have employed five different pre-trained CNN models as our feature extractors and extracted 1024 features from every feature extractor with transfer learning techniques. Then we applied four different feature selection techniques on those 1024 extracted features, including principal component analysis (PCA), recursive feature elimination (RFE), mutual information gain (MIG), and univariate analysis, respectively, to select the significant features. Different numbers of features (e.g., 100, 200…, 900) are chosen for each feature selector before being fed into the classifier to determine the optimal number of features. Five different ML-based classifiers, including decision tree (DT), random forest (RF), XGBoost, multi layer perceptron (MLP), and light gradient-boosting machine (LGBM), respectively, are employed as classifiers. The model was tested for binary and multiclass classification using a variety of performance metrics. To assure real-time prediction for osteosarcoma malignancy using whole slide images, we developed a web application by integrating our proposed model as well.

### 3.1. Dataset

Data on osteosarcomas from the work of Leavey et al. [38] were used in our study. Tumor samples were collected from Children’s Medical Center, Dallas, which consists of 50 patients’s pathology reports of osteosarcoma resection who were treated from 1995 to 2015. A total of 40 WSI (whole slide images) were selected where every WSI represents different sections of the microscopic slide. The WSI represents tumor heterogeneity and response properties as well. At 10X magnification factor, thirty 1024×1024 pixel image tiles from each WSI were randomly selected. After removing irrelevant tiles such as those falling in non-tissue, ink-mark regions, and blurry images, 1144 image tiles were selected from the resulting 1200 tiles. Each image tile is annotated by pathologists in a CSV (Comma Separated Value) file with Tile Identification Number (TIN) and its corresponding classification results. Viable tumor, nontumor, and necrotic tumor are the three main regions used in classification tasks. Among 1144 image tiles 47% (536) are nontumor tiles, 23% (263) are non-viable tumor (necrosis) tiles, and 30% (345) are viable tiles. Figure 2 illustrates sample images of the dataset. For our experiments and investigation, we have taken 80% of the data for training and 20% for testing from the dataset.

### 3.2. Data Pre Processing

The size of original images in our dataset was 1024×1024 pixels. The input for the ImageNet-based pre-trained models is less than or equal to 224×224. If we use transfer learning then the inputs must be suited to the pre-trained model, therefore we have resized all of our input images to 224×224 pixels. We transformed resized images into tensors to work with image intensity values. Then we perform normalization on our images by using this formula:(1)x=(x−mean)/std

Here, *x* stands for the input image that is converted into a tensor representing the pixel intensity, mean is the the average pixel intensity of all images that exist in our dataset and std stands for standard deviation. Normalization enables data distribution that resembles a zero-centered Gaussian curve. By applying normalization, the gradient does not go out of control and makes convergence faster while training. Since we are using RGB images, we have used mean and standard deviation values of 0.5, 0.5, and 0.5 for the red, green, and blue channels, respectively. This resulted in image intensity values in the range of (−1, 1). The preprocessing steps help us to train faster and reduce computational expenses. Figure 3 illustrates our preprocessing process.

### 3.3. Model Selection

#### 3.3.1. Feature Extraction (Deep Learning-Based Feature Extraction)

Feature extraction is the process by which we can extract meaningful information from an input image. DL-based feature extraction mainly uses CNN to extract features from images [39,40,41,42]. In CNN, convolution combined with pooling is utilized as a feature extractor. This study uses five pre-trained CNN models named VGG-16, VGG-19, Xception, ResNet-50, and DenseNet-121 as feature extractors via transfer learning. These pre-trained models were implemented by PyTorch [43] and Keras [44] on ImageNet [45] validation set. The base results of those models on ImageNet validation set is illustrated in Table 1. Here top-1 accuracy is the conventional accuracy (the one with the highest probability), and top-5 accuracy means the model’s top 5 highest probability answers that match with the expected answer (considers a classification is correct of any of the five predictions matches with the ground truth/target label).

#### 3.3.2. VGG-16 and VGG-19

Karen Simonyan and Andrew Zisserman from the University of Oxford proposed VGG Net [46], which took first and second place in the object detection and classification categories of the 2014 ImageNet challenges. VGG Net architecture has two variants in terms of layers, and the variations are VGG-16 and VGG-19.

VGG-16 is a deep CNN model which consists of 16 layers (roughly twice as deep as AlexNet [47]), constructed by stacking uniform convolution, which enhances the network performance. Without using relatively large respective fields in the first convolution layer (e.g., 11×11 with stride 4 in Krizhevsky et al. [47] or 7×7 with stride 2 in Zeiler and Fergus et al. [48]; Sermanet et al. [49]), they used very small (3×3) respective fields throughout the network. A stack of respective small filters (3×3) has been used instead of large (7×7 or 11×11) respective filters because respective small filters make the decision function more discriminative and reduce the number of parameters, allowing for less computational complexity. The 16 in VGG-16 stands for 16 weighted layers known as learnable parameter layers. A total of 21 layers make up VGG-16: 13 convolutional layers, 5 Max Pooling layers, and 3 Dense layers. This model uses ReLU as the activation function following convolution. In the pooling layer, a max pool layer of 2×2 filter with stride 2 has been used throughout the whole architecture. A stack of convolutions is followed by three fully connected layers, the third one having 1000 channels for classification and the first two each have 4096 channels. The dropout value is set to 0.5 for regularization, and Softmax is used as the activation function for classification. The model’s default input tensor size is 224×224 with 3 RGB channels.

VGG-19 is deeper than VGG-16 as it has 19 layers. It has 16 convolution layers, 5 Max Pooling layers, 3 dense layers, which is a total of 24 layers that make up VGG-19. The 3rd, 4th, and 5th convolution of VGG-19 has an extra layer over VGG-16 and the other architectures are the same as VGG-16 i.e., kernel size, stride, padding, pooling, dropout probability, and activation function. It has a much larger number of parameters than VGG-16.

#### 3.3.3. Xception

Francois Chollet introduced Xception from Google research [50]. The architecture is inspired by Inception and entirely based on depthwise separable convolution, where depthwise separable convolution has been used in place of the Inception module [51]. It is based on a solid hypothesis that performs 1×1 convolution to map cross-channel correlations and separately map the spatial correlations of every output channel. This model performs channel-wise spatial convolution followed by a 1×1 convolution to achieve depth-wise separable convolution. The network contains 36 convolutional layers, which form the feature extraction base, and a logistic regression layer is used after the convolutional base for classification. Xception has 14 modules that are made up of 36 convolutional layers, and all of them have a linear residual connection around them except the first and last modules. A global average pooling layer is used at the top layer of this architecture to produce a 1×2048 vector, and several fully-connected layers are kept optional. Here, ReLu has been used as the activation function for non-linearity, and a dropout layer of rate 0.5 has been used before the logistic regression layer in this network. The architecture of this network reduces the number of connections by using depth-wise separable convolution and thus reduces the number of parameters, making it computationally more efficient.

#### 3.3.4. ResNet-50

ResNet was proposed by Kaiming He from Microsoft research [52]. ResNet’s architecture is based on residual learning and is substantially deeper than previous models. Instead of learning unreferenced functions, ResNet explicitly reformulates the layers as learning residual functions. Deeper networks often face a notorious problem of vanishing gradients that hamper convergence [53,54]. ResNet addressed this phenomenon by normalizing initialization and utilizing intermediate normalization layers that enable networks with more layers to converge with backpropagation. ResNet also introduces a deep residual learning approach to overcome this degradation issue. Instead of using a few layers directly, this network uses a residual mapping to fit the underlying mapping by reformulating the residual function F(x):=H(x)−x into F(x)+x (where H(x) is underlying mapping, *x* is input). To formulate F(x)+x, a shortcut connection (skipping one or more layers) has been used. The shortcut connections perform identity mapping, then add their results with the outputs from the stacked layers. ResNet-50 is a variant of ResNet that is a modified version of ResNet-34 with a bottleneck architecture and 50 layers. A bottleneck block contains a stack of 3 layers, which are 1×1, 3×3, and 1×1 convolutions. This architecture uses a batch normalization layer between the convolution and activation layers. ReLU has been used as an activation function, and the dropout layer has not been considered. A global average pooling layer and a fully connected layer of 1000 nodes with softmax are used at the end of this network.

#### 3.3.5. DenseNet-121

DensNet was proposed by Gao Huang et al. [55], and this model enhances feature reuse capabilities based on ResNet in its architecture. It has L(L + 1)/2 direct connections, whereas traditional CNN has L layers with L connections. In DensNet, feature maps are combined using concatenation instead of summing before passing into a layer, and all previous layer’s feature maps are used as input for any specific layer. The Dense Block is the main structure of DensNet, consisting of convolutional layers. DensNet-121 is one of the variants of the DensNet architecture, having a 121-connected convolutional layer with a final output layer. DensNet-121 contains four dense blocks, and there is a transition layer between each dense block. This network’s dense connectivity for $x_{0},x_{1},x_{l-1}$ inputs, where the lth layer receives feature maps from all preceding layers, can be defined as $X_{l}=H_{l}\left(\left[x_{0},x_{1},...,x_{l-1}\right]\right)$. $H_{l}$ is a composite function that contains three operations: batch normalization (BN), rectified linear unit (ReLU), and a 3×3 convolution. Each dense block of DenseNet-121 has two convolutions, 1×1 and 3×3, which are repeated differently in each block. A transition layer contains a 1×1 convolutional layer and an average pooling layer with a stride of 2. Before sending all feature maps to the fully connected layer for classification, this network performs a 7×7 global average pooling layer. This network has fewer parameters than ResNet and is more computationally efficient.

### 3.4. Decision Tree Based Recursive Feature Elimination (DT-RFE)

Recursive feature elimination (RFE) is a wrapper-type feature selection technique that uses different types of machine learning algorithms in its core, and the algorithms help to select features [56,57,58]. RFE fits a model and removes the least significant feature (or features) until the desired(selected) number of features is obtained. The coef or feature importances properties of the model are used to rank the features [59], and RFE attempts to eliminate interdependence and correlation that may exist in the model by recursively eliminating a small number of features per cycle. The goal of RFE is to maximize generalization performance by eliminating the least significant features whose elimination will have the least impact on training errors and select smaller sets of features recursively [60]. There are two major steps that must be taken to implement RFE. Firstly, we need to choose an algorithm (also known as a classifier or estimator) that will give us feature importance, and then we need to specify the number of features we want to select. We have used the decision tree algorithm [61] as our estimator, and different numbers of features were selected for our experiment, i.e., 100, 200, 300, 400, 500, 600, 700, 800, and 900. Decision tree is a tree-based classifier that offers a variety of significance features and performs relatively well. Decision tree algorithms employ information gain to split a node, and for calculating information gain, different criteria can be used [62]. In our experiment two popular criteria have been employed, namely Gini index and entropy to determine which criterion provides better performance based on the data. Mathematically Gini index and entropy can be defined as follows:(2)$$Gini=1-\sum_{i=1}^{n}p^{2}\left(c_{i}\right)$$
(3)$$Entropy=\sum_{i=1}^{n}-p\left(c_{i}\right)log_{2}\left(p\left(c_{i}\right)\right)$$
where $p(c_{i})$ is the probability of class $c_{i}$ in a node. The range of the Gini Index is [0, 0.5], while the range of the entropy is [0, 1]. We have applied multi layer perceptron (MLP) separately on selected features for our final classification.

### 3.5. Multi Layer Perceptron (MLP)

Multi layer perceptron, in short MLP, is a unique variety of an artificial neural network (ANN) [63]. MLP is a feed-forward multilayer network of artificial neurons, and each layer contains a finite number of units (often called neurons) [64,65,66]. Each layer’s unit is connected to each layer’s preceding (and consequently succeeding) unit via a network of connecting lines. Typically, these connections are referred to as links or synapses [67]. Information transmits from one layer to the next layer (thus the term feed-forward). For $x_{1},x_{2},...,x_{n}$ inputs, the model predicts output as $y^{1},y^{2},.....,y^{n}$ with lh hidden nodes or units (*h* is the number of nodes). In this study, MLP model works as follows:The input layer produces output of its *j*th node as $x_{oj}$.The output $x_{ij}$ from each *j*th node of the (i−1)th layer is sent to the *k*th node of the *i*th layer. Then the values of $x_{ij}$ are multiplied by some constants (referred to as weights) $w_{ijk}$, and the resulting products are summed.A shift $b_{ik}$ (referred to as bias) and then a fixed mapping σ (referred to as activation function) are applied to the above sum, and the resulting value represents the output $x_{i+j,k}$ of this *k*th node of the *i*th layer. This can be formulated as follows:
(4)$$x_{i+1,k}=\sigma \left(\sum_{j}w_{ijk}x_{ij}+b_{ik}\right)$$

With the above procedure, for input $x=(x_{1},x_{2},....,x_{n})$, we can write the output $\hat{y}$ of a single hidden layer perceptron model with *q* nodes in the hidden layer as follows:(5)$$\hat{y}=\sum_{i=1}^{q}w_{i}^{2}.\sigma \left(\sum_{j=1}^{n}w_{ij}^{1}x_{j}+b_{i}\right)$$

Here, $w_{ij}^{1}$ is the weight of *j*th unit of the input and *i*th unit in the hidden layer, $b_{i}$ is the bias at the *i*th unit of the hidden layer, and $w_{i}^{2}$ is the weight between the *i*th unit of the hidden layer and the output.

Another step is to determine the values of weights $w_{ij}$ and bias $b_{i}$ in a way that the model behaves well on a given set of inputs and corresponding outputs. This process is called learning or training, and the MLP model uses backpropagation as the basic learning algorithm. Backpropagation is a gradient descent algorithm and mathematically it can be represented as,
(6)$$repeat\;until\;convergence:w_{j}:=w_{j}-\alpha .\frac{\delta }{\delta w_{j}}.J\left(w_{0},w_{1},....,w_{n}\right)$$where $w_{j}$ is weights, α is learning rate, and *J* is the cost function. Cost function basically quantifies the error between the predicted value and the true value of inputs, and mathematically it can be represented as follows:(7)$$J\left(w_{0},w_{1}\right)=\frac{1}{2m}.\sum_{i=1}^{m}{\left(y_{i}-{\hat{y}}_{i}\right)}^{2}$$$y_{i}$ is the actual value, ${\hat{y}}_{i}$ is the predicted value, and *m* is the number of data samples.

Different activation functions can be used in different layers on MLP. In our experiment we have used ReLU as an activation function in the hidden layer. Mathematically ReLU can be defined as follows for input *x*:(8)f(x)=max(0,x)

In the output layer, we have used different activation functions for binary and multiclass classification, respectively. For binary classification we used logistic sigmoid activation function in the output layer, and mathematically it can be defined as follows for input *x*:(9)$$f\left(x\right)=\frac{1}{1+e^{-x}}$$

Additionally, for multi class classification we have used softmax as activation function of the output layer. Softmax can be defined as follows for input *x*:(10)$$\sigma \left(x_{i}\right)=\frac{e_{i}^{x}}{\sum_{k=1}^{N}e^{x_{k}}}$$

In our proposed MLP model, the inputs $\left(x_{1},x_{2},....,x_{n}\right)$ have 1024 features after extracting the feature, so there are 1024 nodes for each input and 100 nodes in the hidden layer. The output layer contains 2 nodes for binary classification and 3 nodes (as we have 3 classes) for multiclass classification. We have used ReLU as the activation function in the hidden layer, logistic sigmoid as the activation function for binary classification and softmax for multiclass classification in the output layer. Furthermore, Adam was used as an optimizer, and it is stochastic gradient-based. $beta_{1}=0.9$, $beta_{2}=0.999$, $epsilon=1\times 10^{-8}$ has been used for the Adam optimizer. We applied L2 regularization with a value of α=0.0001 with a learning rate of 0.001; we trained our model for 200 epochs.

## 4. Experimental Results

The experiments were conducted in the Google Colaboratory environment that includes the NVIDIA Tesla K80 graphics card, 12.68 GB RAM, and 107.72 GB disk space. To implement our proposed model, the Python 3.8.3 programming language with PyTorch, Keras, Scikit-learn frameworks, and various libraries such as Numpy, Pandas, Matplotlib, etc. has been utilized.

### 4.1. Evaluation

The main objective of our proposed model was to classify the osteosarcoma images into one of the three tumor phases (nontumor, necrosis, and viable tumor) as mentioned in the earlier section. In this study, we employed various performance metrics to evaluate our proposed model. Moreover, the impact of the feature selection technique is also analyzed by a comparison of the results before and after applying it based on various performance metrics. Here, accuracy, ROC curve, specificity, sensitivity (recall), precision, F1 score, Matthews correlation coefficient (MCC), and confusion matrix were all considered in the evaluation. Confusion matrix is a table that describes how well a classification algorithm performs, and it visualizes and summarizes the prediction results for a classification problem. In a confusion matrix where true positive (*TP*) stands for a value that is correctly predicted as positive, true negative (*TN*) stands for a value that is correctly predicted as negative, false positive (*FP*) indicates a value incorrectly predicted as positive, and false negative (*FN*) indicates a value incorrectly predicted as negative. Mathematically all these evaluation metrics can be written as follows:(11)$$Accuracy=(\frac{TP+TN}{TP+TN+FP+FN})\times 100\%$$
(12)$$Sensitivity=\frac{TP}{TP+FN}$$
(13)$$Specificity=\frac{TN}{TN+FP}$$
(14)$$Precision=\frac{TP}{TP+FP}$$
(15)$$F1\;score=\frac{2\times (Precision\times Recall)}{Precision+Recall}$$
(16)$$MCC=\frac{\left(TP\times TN)-(FP\times FN\right)}{\sqrt{\left(TP+FP)\times (TP+FN)\times (TN+FP)\times (TN+FN\right)}}$$

We have also considered ROC curves, which represent two-dimensional charts that are frequently used to evaluate and assess the performance of classifiers. It simply illustrates a classifier’s sensitivity or specificity for all possible classification thresholds and indicates how effectively the model can distinguish between different categories. The true positive rate is plotted on the *y*-axis and the false positive rate is plotted in the *x*-axis in this graph, and an AUC close to 1 implies a predicted model does well at class label separability, while an AUC close to 0 indicates a poor predicted model.

### 4.2. Feature Extractor and Classifier Selection

To determine the best feature extractor, we employed five different CNN models, including VGG-16, VGG-19, ResNet-50, DenseNet-121, and Xception. Here, we considered a transfer learning approach rather than training a CNN model from the scratch where pre-trained weights of those models are utilized. The Fully Connected Layer (used as the classifier of a model) of every CNN model was discarded and replaced with five different classifiers based on the ML algorithm. DT, RF, XGBoost, LGBM, and MLP are the algorithms that have been used as classifiers individually with every CNN. The purpose of this experiment with a combinational approach is to investigate and compare the performance of each feature extractor with different ML classifiers. This experiment also allows us to determine the best classifier among all of the ML based classifiers mentioned earlier in this section. A dataset containing tumor samples from patients with osteosarcoma is used to evaluate each combined model; a description of this dataset is given above (see Section 3.1). The investigation was evaluated on the test set, and from this investigation, we select the best feature extractor and the best classifier, as well as the best combination based on their performance. Table 2 represents the experimental results of every combination.

From the experimental results shown in Table 2, we can see that DenseNet-121 combined with all five classifiers achieved the highest average accuracy of 86.16% among all of the feature extractors. VGG-16 obtained the lowest average accuracy among all of them, which is 75.7%. The three other feature extractors, including VGG-19, Xception, and ResNet-50, obtained an average accuracy of 78.44%, 76.86%, and 80.26%, respectively, which are 7.72%, 9.3%, and 5.9% lower than DenseNet-121, respectively. The highest average AUC score is also achieved by DenseNet-121, which is 92.84%. The average AUC scores of DenseNet-121 are 4.02%, 1.96%, 2.8%, and 2.48% higher than the VGG-16, VGG-19, Xception, and ResNet-50 models, respectively. This extractor also achieved the highest average score for other evaluation metrics, including MCC, specificity, and sensitivity, which are 80.58%, 92.7%, and 86.68%, respectively. The average MCC scores of DenseNet-121 are 18.08%, 13.76%, 17.6%, and 11.24% higher than the VGG-16, VGG-19, Xception, and ResNet-50 models, respectively. The specificity and sensitivity are also 8.1%, 5.7%, 7.02%, and 4.16% and 14.46%, 10.88%, 13.68%, and 9.1% higher than the other four feature extractors. When compared to other extractors, these differences in results are quite significant, and DenseNet-121 achieved the highest score in every evaluation metric aspect. DenseNet-121 also achieved the third highest top-5 accuracy of 91.97% on the ImageNet validation dataset and used much fewer parameters than others among all of the mentioned feature extractors (a descriptive overview is provided in Section 3). In our case, DenseNet-121 outperforms all other CNN models on the test dataset for every evaluation metric. Therefore, we have chosen DenseNet-121 as our feature extractor.

From Table 2, we can also see that the MLP classifier achieved the highest average accuracy of 88.68%, which is 16.66%, 14.98%, 4.9%, and 9.44% higher than four other classifiers, including DT, RF, XGBoost, and LGBM, respectively. The MLP classifier also achieved the highest average AUC score of 96.98%, which is 18.44%, 6.16%, 3.3%, and 4.06% higher than the other four mentioned classifiers. The DT, RF, XGBoost, and LGBM obtained average MCC scores of 55.86%, 61.06%, 74.6%, and 68.08%, respectively, while MLP achieved the highest average score of 82.62%. The MCC scores of the other four classifiers are lower than the MLP classifier. The MLP classifier also achieved the highest average score for other evaluation metrics, including specificity and sensitivity, which are 93.9% and 87.8%, respectively. The second-highest average scores for ACC, AUC, MCC, SP, and SN are obtained by the XGBoost classifier, which is 4.9%, 3.3%, 8.02%, 3.24%, and 5.78% lower compared with the MLP classifier; the differences are quite significant. From our investigation, we found that the MLP classifier outperformed all other classifiers that we used in our experiment on our test data. These findings allow us to select MLP as the best classifier. Figure 4 illustrates the ROC-AUC curve for each feature extractor combined with five different classifiers.

The experimental results and our investigation also illustrate that the MLP classifier combined with every feature extractor achieved the highest ACC, AUC, MCC, SP, and SN scores among all other combinations. On the other hand, every classifier combined with DenseNet-121 obtained the highest scores compared with other feature extractors. Finally, we can see that DenseNet-121 combined with the MLP classifier achieved the highest accuracy of 93.4%, which is much higher than all other combinations. By this finding, we have chosen DensNet-121 as a feature extractor and MLP as a classifier, and we have applied this combination to develop our proposed model.

### 4.3. Optimizer Algorithms, Loss, and Convergence Analysis of MLP

The effectiveness and efficiency of optimization algorithms have a significant impact on the implementation of ML models. They generate gradients and try to minimize the loss function that leads to more accurate results. There are many different optimization algorithms that can be implemented to minimize loss in a ML or DL model for supervised, unsupervised, semi-supervised, and reinforcement learning. In our study, three different optimization algorithms have been employed to determine which optimization algorithm works better on our data for MLP, and those are named Stochastic Gradient Descent (SGD), Adaptive Moment Estimation (Adam), and Limited-memory BFGS (Lbfgs), respectively. Table 3 represents the optimization algorithms performance on our data with learning rate, number iteration to convergence, loss, and execution time for both multiclass and binary class classification.

Our experimental results show that SGD takes a higher number of iterations to converge than Adam and Lbfgs. After a several number of experiments we found that SGD takes around 900 iterations to convergence while Adam and Lbfgs takes 500 and 300 iterations, respectively. To investigate the loss value and execution time for each optimization algorithm, we set a certain number of iterations for each of them for both multiclass and binary classification with an initial learning rate of 0.001. The experimental results indicate that Adam and Lbfgs produce both lower loss value and execution time than SGD. In multiclass classification, Adam, Lbfgs, and SGD produce loss values of 0.002621, 0.020567, and 0.000057 with execution times of 3.98 s, 57.15 s, and 2.40 s, respectively. Binary classification’s experimental results also indicate that SGD takes higher execution time, loss value, and number of iteration than Adam and Lbfgs. This investigation motivates us to use Adam as our optimization algorithm for our proposed model as it is more computationally efficient and produces less loss value. Furthermore, we plotted the optimizer analysis for both multiclass and binary class classification based on the no. of iteration, loss, and execution time that is illustrated in Figure 5.

### 4.4. Impact of Feature Selection Techniques

Feature selection (FS) is an effective strategy for choosing the most appropriate feature subset in pattern recognition and medical image processing. This technique helps us eliminate irrelevant features that allow us to build a more straightforward and faster model with higher prediction capability. In recent studies, various feature selection techniques have been widely used in medical image processing and bioinformatics. To investigate the effectiveness of FS techniques, we employed four different feature selectors, namely PCA, RFE, MIG, and univariate, to determine the most effective one that works best in our dataset. We evaluate our proposed model on the test set, which contains 224 images belonging to three different classes. Here, 100, 200, 300, 400, 500, 600, 700, 800, and 900 features were selected individually for each FS technique to determine which number of features yielded the best prediction scores across all evaluation metrics. Both multiclass and binary-class classifications were performed to ensure each FS technique’s impact. Table 4 illustrates the data samples distributed for different classification tasks.

Table 4 shows that we performed a single multiclass classification and four different binary class classifications. Necrosis and viable samples were combined into a class for nontumor versus necrosis + viable binary classification and added nontumor as another class. This binary classification aims to investigate how well our proposed model can classify a tumorous and a nontumorous sample. We also conducted three class-specific binary classifications to ensure that our proposed model can discriminate between two classes in all possible combinations. Initially, multiclass and binary class classification were performed without FS techniques where applied the MLP classifier to 1024 features extracted from DenseNet-121. The results are illustrated in Table 5.

The experimental results of four different feature selection techniques that we used to classify osteosarcoma malignancy for both multiclass and binary class classification are shown in Table 6. This table includes the experimental results for those feature dimensions that achieved the highest performance.

From Table 6, we can see that the average accuracy of five different classification tasks without FS technique is 94.84%, which is 1.58%, 1.88%, 0.5%, and 1.76% lower than the average accuracy of PCA, RFE, MIG, and univariate. This observation shows that FS techniques can improve prediction results significantly on our dataset. The four FS techniques, including PCA, RFE, MIG, and univariate, obtained an average accuracy of 96.42%, 96.72%, 96.34%, and 96.60%, respectively. Among all of the FS techniques, RFE achieved the highest average accuracy, which is 0.30%, 0.38%, and 0.12% higher than PCA, RFE, MIG, and univariate, respectively. The RFE also achieved the highest average score for MCC, precision, and sensitivity, and those scores are 93.56%, 96.48%, and 96.26%, respectively. The highest average AUC achieved by the MIG technique is 98.72%. The RFE obtained an average AUC score of 98.62%, which is slightly lower than the MIG (0.1%). The specificity and F1 scores are more or less the same for every FS technique. The experimental results show that some FS techniques achieved the highest average accuracy on a single evaluation metric, some prediction results have a slight difference from each other, and some predictions are the same for all of the techniques. However, based on various evaluation metrics, we discovered that the RFE FS technique consistently outperformed all of the others. We also investigate the performance based on the number of features. PCA, RFE, MIG, and univariate used an average of 380, 400, 640, and 420, respectively, to obtain their best prediction results. Though PCA uses a smaller average number of features for the best prediction, it does not provide better results than RFE. From the experimental results, we can see that for multiclass classification, it uses only 100 features to predict the best results, but its accuracy is 0.9% lower than the RFE technique. PCA also uses a higher number of features than RFE in all four binary classifications, and its performance is also significantly lower. RFE and univariate use a higher average number of features than RFE to obtain their best prediction results. In the binary classification, we can see that RFE uses a smaller number of features than all other FS techniques except in the nontumor versus viable tumor classification. We implemented DT-based RFE using the Scikit-Learn (sk-learn) library, where DT has been used as an estimator that has been discussed in Section 3.2. As this library offers two different criteria for the DT estimator, we also analyze its criterion based on the execution time for a more sophisticated version of DT-RFE that works on our dataset. Table 7 represents the execution time during the experiment using Gini and entropy criterion.

In this table, it is shown that the entropy criterion takes much more time than the Gini index to select the most prominent features for both multiclass and binary class classification. In multiclass classification, the Gini index takes around 5.23 min while entropy takes 8.41 min, and in other binary classifications including nontumor vs. necrosis and viable, nontumor vs. necrosis, necrosis vs. viable, and nontumor vs. viable entropy takes more execution time than the Gini index. Less execution time is more computationally efficient, which motivates us to use the Gini index as our criterion for the decision tree estimator in the RFE feature selection technique. We also plotted a violin plot for our selected feature selector that is shown in Figure 6.

In addition to our study, we applied gradient-weighted class activation mapping (Grad-CAM) [68] to further analyze and explain the feature extractor of our proposed model. All convolutional layers in a CNN retain their respective spatial information that is lost in the FC layer. Grad-CAM uses the gradient information flowing into the last convolutional layer of the CNN to assign importance values to each neuron for a particular decision of interest, as this last layer contains high-level semantics and detailed spatial information of an input image. This method computes the gradient of the target class score with respect to the last layer’s feature maps, weights computed gradients by average pooling for the importance of each feature map channel, and finally combines weighted gradients to generate a heat map that illustrates the feature importance. We have overlayed this generated heatmap with input images to obtain the Grad-CAM images. Figure 7 represents some sample Grad-CAM images from our dataset that have been utilized as input images. By visualizing those images, we can investigate the decision-making process of our proposed model, where the red regions are the affected areas considered by the model.

### 4.5. Web Application for Osteosarcoma Classification

A web application is developed using our proposed model with real-time validation to classify osteosarcoma using whole slide images as input. A modern, fast (high-performance) web framework named FastAPI [69] has been used to develop our web application on the Python 3.8.3 version. FastAPI is used for building APIs and backend development, and we have used HTML, CSS, and JavaScript for frontend development. The workflow of our web application is given in Figure 8.

Initially, the user needs to select an image from the user interface as input to see the classification result. The input image will be preprocessed based on what has been used in our training phase, then fed into the pre-trained CNN to extract features. The features will be scaled through the loaded feature scaler and fed into the RFE feature selector. The selected features will be fed into the loaded classifier for prediction. We have performed all five classifications including multiclass and binary with saved classifiers, then max-voted the predicted class from all classifiers, and selected the most frequent class as the predicted class. This max-voting process ensures the reliability of our model for web applications.

After developing the web application, it has been deployed on a cloud platform as a service named Render [70]. Render provides a publicly accessible URL by which any user can access web applications that have been deployed on this platform. The home page that takes inputs (Figure 9a) and the output page (Figure 9b–d) are presented in Figure 9. Users need to click on the select an image file box to upload an image, which prompts up their local store where they can select the input image. By clicking on the submit button, the user will be able to see the output results for a given image including predicted class, class probability, and inference time. Some random images are given as input to evaluate the robustness of our proposed model using the web applications for real-time validation, and the result is shown in Figure 9.

### 4.6. Comparison with Existing Models

In this section, we compared our proposed hybrid model with existing state-of-the-art models that were developed to identify osteosarcoma malignancy on the dataset. To ensure the effectiveness and robustness of the proposed model, the comparison was performed with different performance metrics including accuracy, precision, recall, and F1 score. Our proposed model has been compared with Mishra, Rashika et al. [71], Mishra et al. [10], Arunachalam, Harish Babu et al. [33], Nabid et al. [5], and Anisuzzaman et al. [4] in terms of mentioned performance metrics listed in Table 8.

As shown in Table 8, Mishra, Rashika et al. [71], Mishra et al. [10], and Nabid et al. [5] reported low accuracy, precision, recall, and F1 score. Arunachalam, Harish Babu et al. [33] obtained the highest accuracy of 89.9% for the ML approach by using SVM. They also employed deep learning approaches where they obtained the highest accuracy of 91.2% for tiles (WSI). Both of their accuracy results refer to class-specific accuracy, and they did not talk about precision, recall, and F1 score. Anisuzzaman et al. [4] achieved the highest accuracy of 96% for binary class classification using VGG-19 via transfer learning; this is 3.4% lower than ours. Furthermore, the model’s precision, recall, and F1 score are 4%, 5%, and 4%, respectively, which are all lower than our proposed model. Our proposed model achieved the highest accuracy of 99.4%, which is higher than all existing models. Furthermore, the precision, recall, and F1 score of our model are 0.99, 1.00, and 0.99, which are also higher than all existing models. From this investigation and comparison, we can clearly see that our proposed model outperforms all the existing models in the literature so far based on various evaluation metrics. The robustness and high performance of the proposed model are achieved due to the techniques we have developed and implemented for classifying osteosarcoma malignancy. Initially, normalization techniques were performed in the preprocessing step to enable our model to learn faster during the training phase with a zero-centered Gaussian distribution of data. We have also explored various CNN models and ML classifiers to select the best feature extractor and classifier by conducting huge experiments on the dataset. A total of 25 different combinations of CNN models and ML classifiers were evaluated with different parameter settings to determine the best integration (as shown in Table 2). Furthermore, an optimizer analysis was conducted for the MLP classifier to ensure better optimization by selecting the most suitable optimizer for classifying osteosarcoma that demonstrates improved convergence time and loss. In addition to the above techniques, we have investigated various feature selection techniques where DT-based RFE is selected based on performance. We also analyzed the impact of RFE’s criterion concerning time and the number of selected features, aiming for enhanced performance as well as reduced computational cost. Moreover, the integration of CNN and ML with feature selectors leverages the advantages of each approach. This integration makes our proposed model more robust and outperforms all the existing state-of-the-art models to classify osteosarcoma on this dataset.

## 5. Conclusions

Classification of osteosarcoma malignancy using histological biopsy by pathologists is quite challenging, tedious, and time-consuming. In this paper, we proposed a hybrid model that combines DL and ML to classify osteosarcoma malignancy that will help pathologists with a computer-aided system. First, it extracts relevant features from whole slide images using DenseNet- 121, then performs feature selection using DT-RFE to select the most significant features, and, finally, the MLP classifier is applied to those features chosen for osteosarcoma classification. However, we utilized transfer learning (pre-trained CNN models) for feature extraction rather than building a CNN model from scratch, as it requires a large amount of data and a higher training time. Feature selection techniques have been applied in our model to reduce feature dimensions. Transfer learning, DenseNet- 121, and the feature selection DT-RFE technique reduce computational costs and make our model faster. Moreover, from the five well-known ML algorithms, we selected MLP for classification as the best-performing algorithm based on the performance of our dataset. The experimental results illustrate that our proposed model has higher prediction performance than existing state-of-the-art models developed for osteosarcoma malignancy classification on the same dataset. We also developed a web application of our proposed model that can be used in clinics for early diagnosis of osteosarcoma. After applying feature selection techniques, the accuracy has increased 1.8% for multiclass classification. For binary classification, it has been increased by 2.2%, 1.2%, 2.3%, and 1.6% for nontumor vs. necrosis + viable, nontumor vs. necrosis, nontumor vs. viable, and necrosis vs. viable, respectively. We believe our proposed hybrid model is not only applicable to osteosarcoma classification, but also it can be applied to other histopathological image classifications. In the future, we plan to integrate uncertainty mining and pLOF techniques into our model to make our predicted results more trustworthy.

## Figures and Tables

**Figure 1 diagnostics-13-02106-f001:**
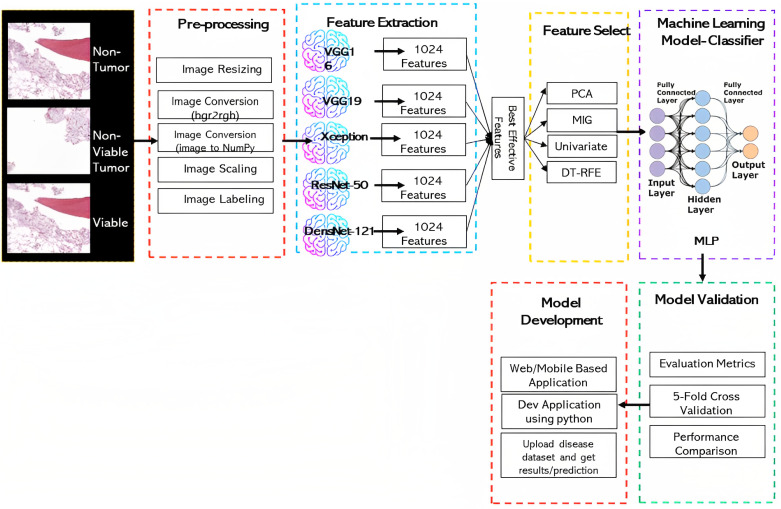
Proposed methodology.

**Figure 2 diagnostics-13-02106-f002:**
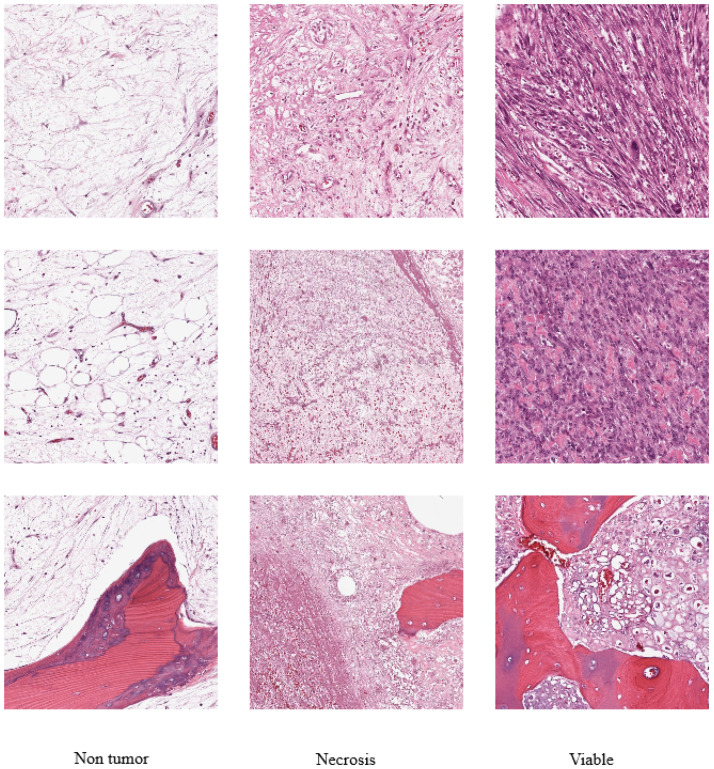
Sample images from our dataset [38].

**Figure 3 diagnostics-13-02106-f003:**
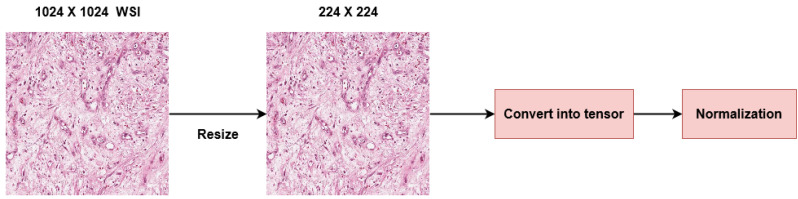
Data preprocessing stages where images were resized then performed normalization.

**Figure 4 diagnostics-13-02106-f004:**
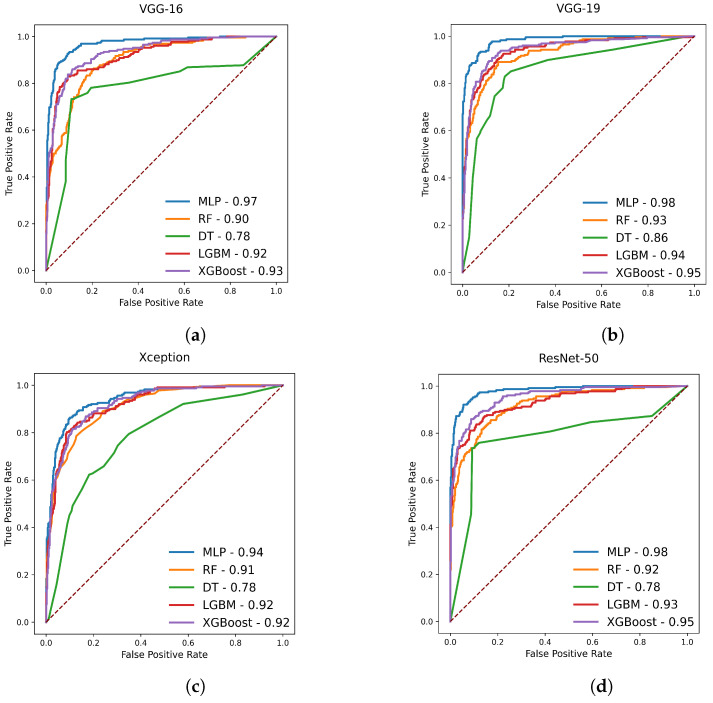
ROC-AUC for five feature extractor. (**a**) ROC-AUC for VGG-16; (**b**) ROC-AUC for VGG-19; (**c**) ROC-AUC for Xception; (**d**) ROC-AUC for ResNet-50; (**e**) ROC-AUC for DenseNet-121.

**Figure 5 diagnostics-13-02106-f005:**
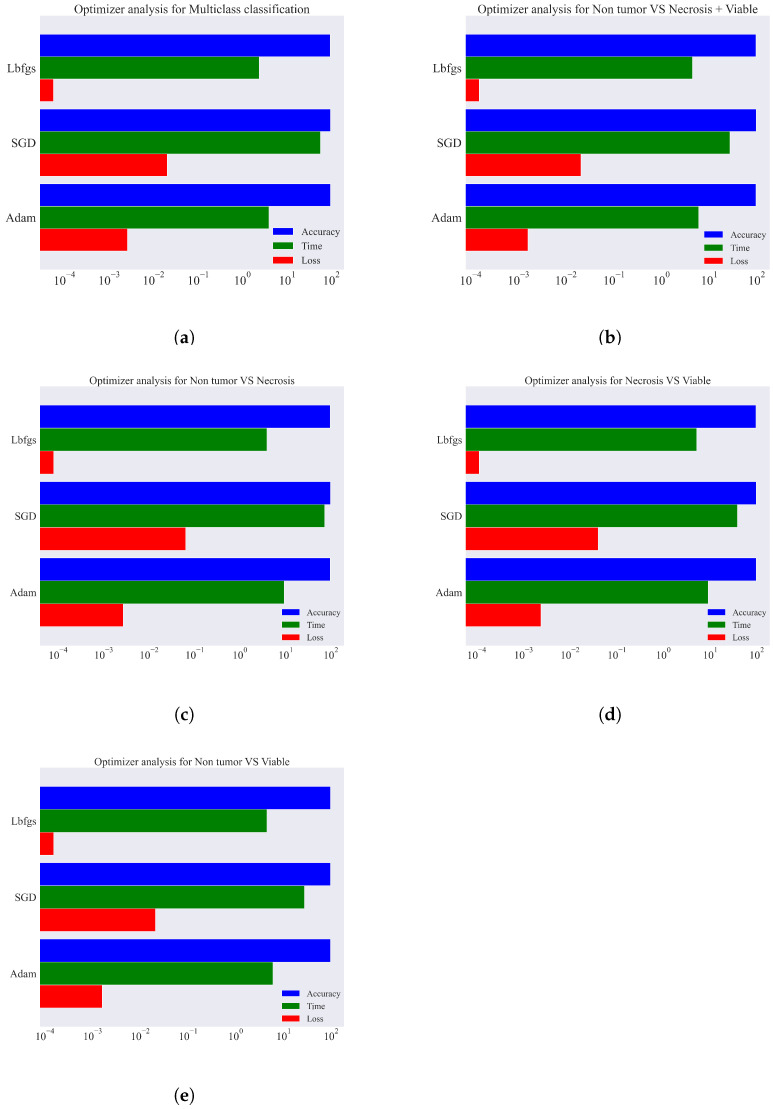
Optimizer analysis for both multiclass and binary class classification based on no. of iteration, loss, and execution time. (**a**) Analysis for multiclass. (**b**) Analysis for NT vs. Nec. + Via. (**c**) Analysis for NT vs. Nec. (**d**) Analysis for Nec. vs. Via. (**e**) Analysis for NT vs. Via.

**Figure 6 diagnostics-13-02106-f006:**
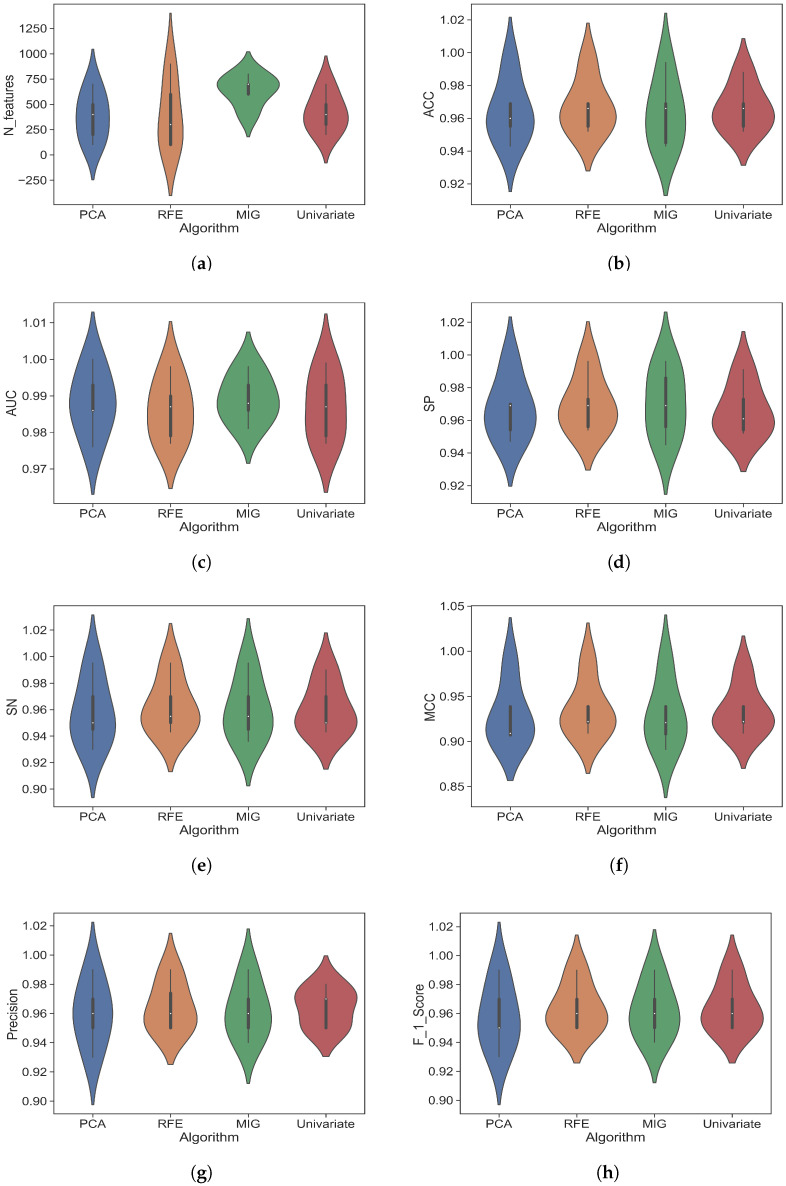
Violin plot the experimental results after applying 4 different feature selection techniques. (**a**) For number of features. (**b**) For accuracy. (**c**) For AUC. (**d**) For specificity. (**e**) For sensitivity. (**f**) For Matthew’s correlation coefficient. (**g**) For precision. (**h**) For F1 score.

**Figure 7 diagnostics-13-02106-f007:**
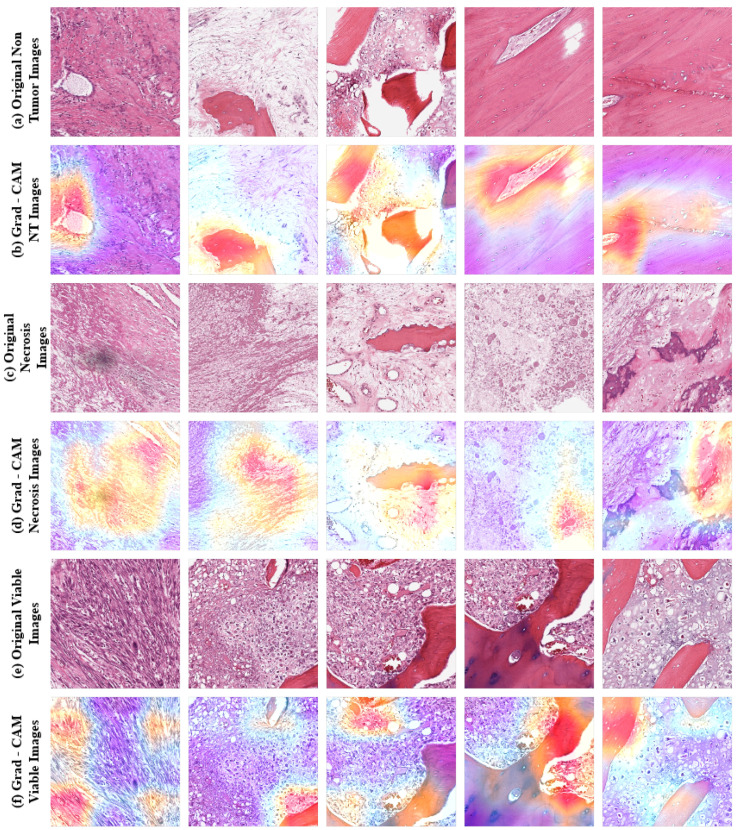
Example of some input images and their Grad-CAM images generated by proposed model, where red regions indicated the affected area.

**Figure 8 diagnostics-13-02106-f008:**
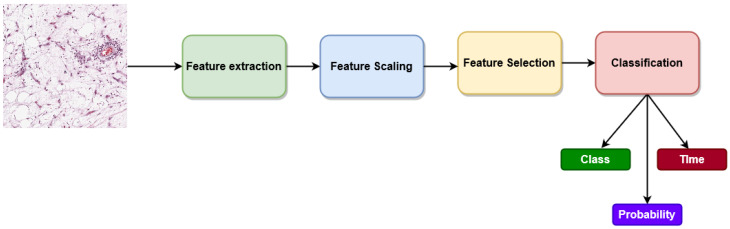
Workflow diagram of our developed web application.

**Figure 9 diagnostics-13-02106-f009:**
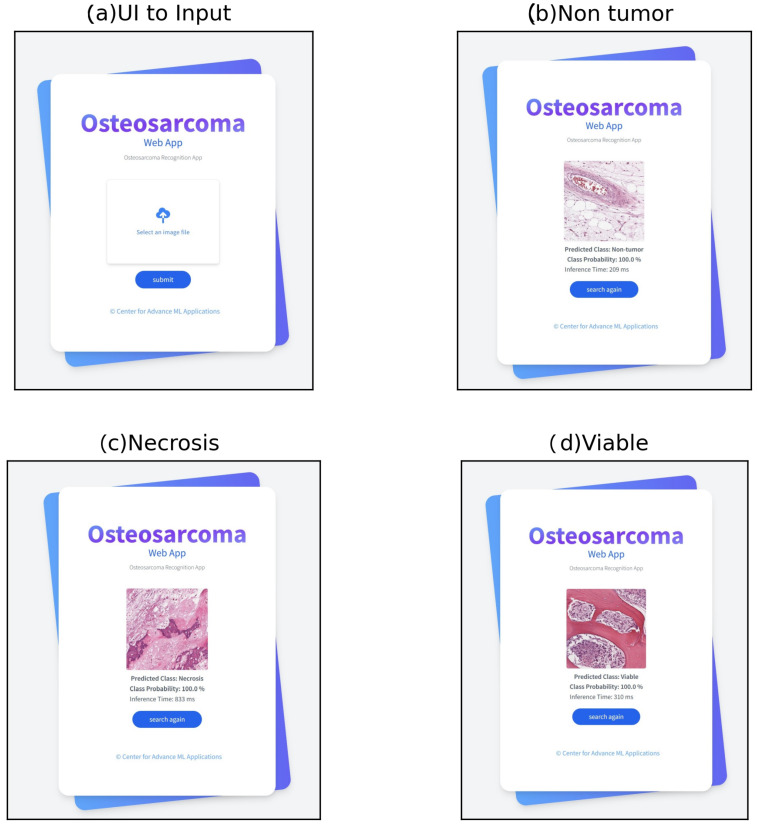
Sampleinput and output from our web application.

**Table 1 diagnostics-13-02106-t001:** Parameters and accuracy of pre-trained models (acc. stands for accuracy).

CNN Model	Parameters	Top-1 acc.	Top-5 acc.
VGG-16	138.4M	71.592%	90.382%
VGG-19	143.7M	72.376%	90.876%
Xception	22.9M	79.0%	94.5%
ResNet-50	25.6M	76.13%	92.862%
DenseNet-121	8M	74.434%	91.972%

**Table 2 diagnostics-13-02106-t002:** Experimental results of the performance of feature extractor with different classifier.

Extractor	Classifier	ACC	AUC	MCC	SP	SN
VGG-16	Decision Tree	0.680	0.768	0.497	0.802	0.650
	Random Forest	0.627	0.877	0.437	0.743	0.580
	XGBoost	0.811	0.921	0.705	0.890	0.791
	LGBM	0.759	0.901	0.631	0.845	0.700
	MLP	0.908	0.974	0.855	0.950	0.890
VGG-19	Decision Tree	0.684	0.814	0.505	0.805	0.660
	Random Forest	0.737	0.912	0.602	0.836	0.700
	XGBoost	0.838	0.930	0.747	0.906	0.820
	LGBM	0.768	0.913	0.649	0.857	0.720
	MLP	0.895	0.975	0.838	0.946	0.890
Xception	Decision Tree	0.636	0.786	0.418	0.766	0.610
	Random Forest	0.781	0.916	0.649	0.861	0.720
	XGBoost	0.803	0.928	0.684	0.880	0.760
	LGBM	0.803	0.934	0.685	0.882	0.760
	MLP	0.820	0.938	0.713	0.895	0.800
Resnet-50	Decision Tree	0.776	0.762	0.648	0.871	0.739
	Random Forest	0.715	0.892	0.553	0.823	0.680
	XGBoost	0.842	0.948	0.755	0.916	0.830
	LGBM	0.803	0.943	0.699	0.879	0.760
	MLP	0.877	0.973	0.812	0.938	0.870
DenseNet-121	Decision Tree	0.825	0.797	0.725	0.901	0.820
	Random Forest	0.825	0.944	0.812	0.930	0.860
	XGBoost	0.895	0.957	0.839	0.941	0.900
	LGBM	0.829	0.955	0.740	0.897	0.814
	MLP	0.934	0.989	0.913	0.966	0.940

**Table 3 diagnostics-13-02106-t003:** Experimental results of optimizer analysis.

Classification	Optimizer Name	LR	Iteration	Loss	Time (s)
Multi class	Adam	0.001	500	0.002621	3.98
	SGD	0.001	900	0.020567	57.15
	Lbfgs	0.001	300	0.000057	2.40
Nontumor vs. Necrosis + Viable	Adam	0.001	500	0.001610	6.09
	SGD	0.001	900	0.020836	27.80
	Lbfgs	0.001	300	0.000153	4.54
Nontumor vs. Necrosis	Adam	0.001	500	0.002714	9.42
	SGD	0.001	900	0.064409	73.59
	Lbfgs	0.001	300	0.000080	3.95
Necrosis vs. Viable	Adam	0.001	500	0.002406	9.18
	SGD	0.001	900	0.040208	38.84
	Lbfgs	0.001	300	0.000114	5.22
Nontumor vs. Viable	Adam	0.001	500	0.003227	4.94
	SGD	0.001	900	0.051609	40.33
	Lbfgs	0.001	300	0.000098	3.821

**Table 4 diagnostics-13-02106-t004:** Number of samples in each class in our test dataset (our evaluation based on this data). NT = Nontumor, Nec. = Necrosis, Via. = Viable.

Classification	NT	Nec.	Via.	Nec. + Via.	Total
Multiclass	118	56	54	-	228
NT vs. Nec. + Via.	118	-	-	110	228
NT vs. Nec.	118	56	-	-	174
NT vs. Via.	118	-	54	-	172
Nec. vs. Via.	-	56	54	-	110

**Table 5 diagnostics-13-02106-t005:** Experimental classification results without feature selection.

Classification	No. of Features	ACC	AUC
Multiclass	1024	0.934	0.985
NT vs. Nec. + Via.	1024	0.947	0.993
NT vs. Nec.	1024	0.954	0.983
NT vs. Via.	1024	0.971	0.997
Nec. vs. Via.	1024	0.936	0.978

**Table 6 diagnostics-13-02106-t006:** Experimental results of feature selection using different feature selection techniques.

Classification	Algorithm	No. of Feat.	ACC	AUC	SP	SN	MCC	Prec.	F1 Score
Multi class	PCA	100	0.943	0.986	0.970	0.930	0.907	0.930	0.930
	RFE	900	0.952	0.987	0.973	0.943	0.922	0.950	0.950
	MIG	400	0.943	0.986	0.986	0.936	0.908	0.940	0.940
	Univariate	700	0.952	0.987	0.973	0.943	0.922	0.950	0.950
Nontumor vs. Nec. + Via.	PCA	200	0.969	0.993	0.969	0.970	0.939	0.970	0.970
	RFE	100	0.969	0.990	0.969	0.970	0.939	0.974	0.970
	MIG	600	0.969	0.993	0.969	0.970	0.939	0.970	0.970
	Univariate	200	0.969	0.993	0.961	0.970	0.939	0.970	0.970
Nontumor vs. Necrosis	PCA	500	0.960	0.976	0.947	0.945	0.907	0.960	0.950
	RFE	100	0.966	0.979	0.956	0.955	0.921	0.960	0.960
	MIG	700	0.966	0.988	0.956	0.955	0.921	0.960	0.960
	Univariate	500	0.966	0.979	0.952	0.950	0.921	0.970	0.960
Nontumor vs. Viable	PCA	700	0.994	1.000	0.996	0.995	0.987	0.990	0.990
	RFE	600	0.994	0.998	0.996	0.995	0.987	0.990	0.990
	MIG	700	0.994	0.998	0.996	0.995	0.987	0.990	0.990
	Univariate	300	0.988	0.999	0.991	0.990	0.978	0.980	0.990
Necrosis vs. Viable	PCA	400	0.955	0.986	0.954	0.950	0.909	0.950	0.950
	RFE	300	0.955	0.977	0.954	0.950	0.909	0.950	0.950
	MIG	800	0.945	0.981	0.945	0.945	0.891	0.950	0.950
	Univariate	400	0.955	0.977	0.954	0.950	0.909	0.950	0.950

**Table 7 diagnostics-13-02106-t007:** Experimental results of DT-RFE criteria analysis.

Classification	Criterion	Exec. Time
Multi class	Gini (500)	5.23 m
	entropy	8.41 m
NT vs. Nec. + Via.	Gini (100)	8.61 m
	entropy	12.18 m
NT vs. Nec.	Gini (600)	3.35 m
	entropy	4.03 m
Nec. vs. Via.	Gini (300)	4.95 m
	entropy	5.46 m
NT. vs. Via.	Gini (600)	1.93 m
	entropy	2.24 m

**Table 8 diagnostics-13-02106-t008:** Comparison of our proposed model with existing state-of-the-art models.

Article Author	Highest Acc. (%)	Precision	Recall	F1 Score
Mishra, Rashika et al. [71]	84	0.89	0.84	0.86
Mishra et al. [10]	92	0.97	0.94	0.95
Arunachalam, Harish Babu et al. [33]	89.9/91.2	-	-	-
Nabid et al. [5]	89	0.88	0.89	0.88
Anisuzzaman et al. [4]	96	0.95	0.95	0.95
**Proposed model**	**99.4**	**0.99**	**1.00**	**0.99**

## Data Availability

All developed models and implemented codes are available at: https://github.com/taareek/osteosarcoma_classification.

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
