# Peer review of "A Novel Hybrid Approach for Classifying Osteosarcoma Using Deep Feature Extraction and Multilayer Perceptron"

_diagnostics, 2023, doi:10.3390/diagnostics13122106_

Round 1
Reviewer 1 Report
Thank you for your effort and interesting work.
I have some comments that must be considered in the modified manuscript.
-----------------------------------------------------------------------------------
The paper introduces a hybrid framework by merging different types of CNN-based architectures with a multilayer perceptron (MLP) algorithm on the whole slide images dataset, to improve the efficiency of three types of osteosarcoma tumor.
-------------------------------
1) At the end of (Introduction), authors explained the hybrid framework used. But, it is needed to clarify its novelty (you already wrote about previous work...so, just mention the difference between your framework and the previous one).
2) At the end of (Introduction), you need a paragraph for paper organization.
3) Work procedure is clarified in Fig. 1. Thank you.
4) Sample data set shown in Fig. 2 needs a reference (or: source of data).
5) The tensor equation: Equation (1) needs more clarification: what is x and what is mean????
6) Before presenting Table 1, please explain how did you get the accuracy of Table 1.
7) Equation (5) is a definition (universal)? or: it needs a reference? Same also for Eq. (6).
8) Table 2 compares the ML parameters using different techniques, which is good, from which one can choose the most suitable technique. But, the question is: why you did not compare with a previously published work?, which would be better to judge your results. (same also is Table 3 + Table 6).
9) Figure 5 is very interesting. Please use larger fonts for numbering both axes.
10) Table 8 answers some of my comments. Please put Table caption up not below table.
11) References are up-to-date but nothing is used in 2023.
Reviewer 2 Report
In this manuscript, the authors presented a hybrid framework that combines different types of convolutional neural network (CNN) architectures with a multilayer perceptron (MLP) algorithm. They utilized a dataset of whole slide images (WSI) and applied various pre-processing techniques to the images. Overall, the article presents good work in each section. The Major improvements are needed in the experiment results section for this manuscript.
1. In section 4.6 Comparison with existing models, the authors compare their model with existing methods and find that it outperforms them. However, the review papers also presented high accuracy in their work. My suggestion is that the authors should give a clear contribution, more explanation and discussion of why the proposed hybrid model performed better results.
2. Previous papers in image classification using deep learning presented the results of model explanation, such as using the Shapley Values method, and Grad-CAM to visualize class activation maps. The manuscript experimented with different feature selection methods. Could they explain the critical features for classifying three types of osteosarcoma tumor? I think the model explanation could be helpful in supporting doctors in osteosarcoma diagnosis in clinics.
Author Response
In this manuscript, the authors presented a hybrid framework that combines different types of convolutional neural network (CNN) architectures with a multilayer perceptron (MLP) algorithm. They utilized a dataset of whole slide images (WSI) and applied various pre-processing techniques to the images. Overall, the article presents good work in each section. The Major improvements are needed in the experiment results section for this manuscript.
Question-01: In section 4.6 Comparison with existing models, the authors compare their model with existing methods and find that it outperforms them. However, the review papers also presented high accuracy in their work. My suggestion is that the authors should give a clear contribution, more explanation and discussion of why the proposed hybrid model performed better results.
Response-01: Thank you for your suggestion. We have tried to clear our contribution with more explanation and discussion of reason outperformed the proposed hybrid model in the revised version of the paper. (Section 4.6 “Comparison with existing models”, Line No:660-677).
Question-02: Previous papers in image classification using deep learning presented the results of model explanation, such as using the Shapley Values method, and Grad-CAM to visualize class activation maps. The manuscript experimented with different feature selection methods. Could they explain the critical features for classifying three types of osteosarcoma tumor? I think the model explanation could be helpful in supporting doctors in osteosarcoma diagnosis in clinics.
Response-02: Thank you for your suggestion. We have implemented Grad-CAM for our proposed model for a visual explanation. Grad-CAM images will support doctors in osteosarcoma diagnosis in clinics as it represents the affected area using the heatmap. Besides, we have employed Decision Tree based RFE earlier in our proposed framework. DT-RFE recursively selects only those features that have much impact on the final prediction and eliminates the less impactful features. We can say our proposed model can handle critical features for osteosarcoma cancer malignancy. We have done experiments with Grad-CAM and added an explanation in the revised version of the paper, (Figure:7, Line No: 597-611)
Reviewer 3 Report
In the manuscript titled ‘A Novel Hybrid Approach for Classifying Osteosarcoma using Deep Feature Extraction and Multilayer Perceptron’, the authors introduced a hybrid framework for improving the efficiency of osteosarcoma tumor classification by merging different types of CNN-based architectures. The authors should make some revisions to address the following issues:
(1) How is the allocation of datasets for training and testing?
(2) I recommend using N-fold cross validation for performance evaluation.
(3) One algorithm outperforms another one can only be verified by statistical analysis.
(4) Please take care of the format of the manuscript. E.g.: P9, Line326, "Where" should be "where"; P10, Line356&359, there should be no indentation; P22, The caption of Table 8 should be on the top of the table.
The format of the manuscript must be polished.
Author Response
In the manuscript titled ‘A Novel Hybrid Approach for Classifying Osteosarcoma using Deep Feature Extraction and Multilayer Perceptron’, the authors introduced a hybrid framework for improving the efficiency of osteosarcoma tumor classification by merging different types of CNN-based architectures. The authors should make some revisions to address the following issues:
Question-01: How is the allocation of datasets for training and testing?
Response-01: Thank you for your comment. In this experiment, we considered 80% of the data for training and 20% for testing with the WSI dataset. We have added this information in Section 3.1 Dataset (Line No: 218-219).
Question-02: I recommend using N-fold cross-validation for performance evaluation.
Response-02: Thank you for your comment. We have applied 5-fold cross-validation for performance evaluation, and it has been mentioned in “Abstract” (Line No: 14). All the experimental results presented in this study were done under 5-fold cross-validation.
Question-03: One algorithm outperforms another one can only be verified by statistical analysis.
Response-03: Yes, one algorithm outperforms another one verified by statistical analysis. To ensure that, we have evaluated every algorithm concerning different statistical evaluation metrics such as accuracy, AUC, ROC Curve, specificity, sensitivity (recall), precision, F-1, MCC, and violin plots (Section 4.1 Evaluation) to show the results of the algorithms, feature extractors and feature selectors (Please see Table 2 and 6).
Besides, our model verification we have implemented Grad-CAM to visualize class activation maps which clearly visualize the disease-infected areas of the WSI image dataset (Please see Figure: 7 and Lines: 597-611). Grad-CAM images will support doctors in osteosarcoma diagnosis in clinics as it represents the affected area using the heatmap.
Question-04: Please take care of the format of the manuscript. E.g.: P9, Line326, "Where" should be "where"; P10, Line356&359, there should be no indentation; P22, The caption of Table 8 should be on the top of the table.
Response-04: Thank you so much for your correction, we have addressed the mentioned issues. We have removed the indentation as well as replaced the caption of Table 8 on the top of the Table.
Comments on the Quality of English Language: The format of the manuscript must be polished.
Response: Thank you for your suggestion. English proofreading is done by our English country authors in the revised version of the paper.
Round 2
Reviewer 2 Report
The author has responded the questions and revised the manuscript of the paper.
Reviewer 3 Report
The authors have made revisions according to my suggestions. Now I recommend its publication.